# Hydrophobic chirality amplification in confined water cages

Choong Eui Song[1], Si Joon Park[1], In-Soo Hwang [1], Min Jung Jung[1], So Young Shim[1], Han Yong Bae[1] & Ji Yoon Jung[1]

The manipulation of the transition states of a chemical process is essential to achieve the desired selectivity. In particular, transition states of chemical reactions can be significantly modified in a confined environment. We report a catalytic reaction with remarkable amplification of stereochemical information in a confined water cage. Surprisingly, this amplification is significantly dependent on droplet size. This water-induced chirality amplification stems from the hydrophobic hydration effects, which ensures high proximity of the catalyst and substrates presumably at the transition state, leading to higher enantioselectivity. Flow and batch reactors were evaluated to confirm the generality of this water-induced chirality amplification. Our observation on efficient chiral induction in confined water cages might lead to an understanding of the chirality amplification in the prebiotic era, which is a key feature for the chemical evolution of homochirality.

[1] Department of Chemistry, Sungkyunkwan University, Suwon 16419, Korea. Correspondence and requests for materials should be addressed to C.E.S. (email: s1673@skku.edu)

Asymmetric catalysis provides valuable chiral organic molecules by controlling the enantio-determining transient catalyst-substrate complex. The action of chiral catalysts could be tuned by altering their chemical structure, however, reaction conditions also play a significant role[1]. In principle, an energy difference of ca. 3 kcal mol$^{-1}$ is sufficient to guide an asymmetric reaction pathway towards absolute selectivity (>99% ee) at ambient temperature. This level of energy is comparable to that of a non-covalent weak intermolecular interaction such as hydrogen bonding, π−π interaction, hydrophobic interaction or physical adsorption[1,2]. Therefore, the desired enantioselectivity can theoretically be achieved by altering environmental factors such as the confinement effect[2], high pressure effect[1], and hydrophobic effect[3,4] which affect reaction outcome by generating compact transition states.

In nature, water is used not only as a reaction medium but also as a reaction enforcer in enzymatic processes for biosynthetic reactions to sustain life, by inducing hydrophobic interactions between enzymes and substrates[5]. From the perspective of green chemistry and other scientific efforts to mimic nature, considerable efforts have recently been made to develop asymmetric catalytic reactions in aqueous environments[6–9]. In particular, on-water catalysis[10,11] has recently received considerable attention from scientific communities, due to its sustainability and effectiveness[11–17]. Chemical reactions with negative volumes of activation can be accelerated under on-water conditions due to hydrophobic hydration effects[15–17] (Fig. 1a). Several research groups independently reported that, even hydrogen-bonding promoted catalysis could also be significantly amplified in aqueous environments by the hydrophobic hydration effect[18–22]. Very recently, we demonstrated that this hydrophobic amplification achieved under on-water conditions also enabled to discover new catalytic reactions of otherwise completely unreactive substrates[22] (Fig. 1b). A somewhat related phenomenon is the substantial rate acceleration seen in aqueous microdroplet chemistry[23–25]. Chemical reactions in micrometer diameter droplets can be orders of magnitude faster than their conventional bulk-phase counterparts (Fig. 1a). Note that the reaction rates in microdroplets are usually inversely proportional to the radius of droplets[26]. Moreover, aqueous microdroplet conditions often drive even thermodynamically uphill reactions such as the direct condensation reactions of nucleobases with ribose, producing purine and pyrimidine ribonucleosides which are the building blocks of ribonucleic acid (RNA)[27].

Considering the above-mentioned potential of water as a reaction enforcer, we were intrigued by the possibility of controlling not only the reaction rate but also the enantioselectivity in a confined space of droplets, in which the reactivities and selectivities would differ from those in bulk solutions. In a confined water cage, a hydrophobic catalyst and non-polar substrates would be brought together in a close proximity, consequently constructing sterically more confined transition states, leading to higher stereoselectivities of asymmetric catalysis.

We report our finding that, under on-water conditions, enantioselectivities of a catalytic reaction can be dramatically amplified in the confined hydrophobic cavities of microdroplets surrounded by hydration shell. The observed chirality amplification was further enhanced by decreasing the size of droplets. The effect of the droplet size on the enantioselectivity was quantified by using the biphasic microfluidic technique. The presented experimental data suggest the role of water, not only as a reaction medium in the prebiotic era but also as a key component in controlling transition states in a chiral fashion.

## Results

**Initial findings on water-induced chirality amplification.** To validate our assumptions, we initially examined the Mannich reaction of N-Boc (tert-butyloxycarbonyl) protected imine **1a** with 2,4-pentanedione **2** or dimethyl malonate **4**, using 1 mol% of readily available natural (+)-cinchonine (**CN-1**) as the catalyst at 20 °C as a model reaction (Fig. 1c). Notably, the Schaus group reported that **CN-1** catalyzed the asymmetric Mannich reaction of N-Boc imines with 1,3-dicarbonyl compounds[28,29]. However, to obtain reasonable enantioselectivity, high catalyst loading (10 mol%) and low reaction temperatures (−35 °C) were required. With the reduced catalyst loading (1 mol%) and under room temperature conditions, very poor enantioselectivities were obtained in conventional organic solvents (Fig. 1c and Supplementary Table 1). For example, in CH$_2$Cl$_2$ (Schaus's condition) only 22% ee and 32% ee were obtained for (R)-**3a** and (R)-**5a**, respectively. Other polar solvents such as THF, CH$_3$CN, EtOH, and HCHO gave almost racemic products (1–4% ee). However, the enantioselectivity considerably increased simply by changing the reaction medium from organic solvents to brine in which organic reactants are dispersed, not solubilized (i.e., on-water condition[10]) (55% ee for **3a** and 42% ee for **5a**, see Fig. 1c), probably due to the hydrophobic hydration effect. However, these ee values achieved by employing an on-water condition were still low. As shown in entries 8 and 17 of Supplementary Table 1, brine only catalyzes the reaction to give racemic products (0% ee). On brine, full conversion was observed in the absence of the catalyst within 0.5 h. It was postulated that the interfacial hydrogen bonding or a proton transfer process between the aqueous phase and organic reactants can promote this non-enantioselective process under on-water conditions[30]. Thus, we set out experiments to suppress the water-catalyzed racemic pathway by providing water/oil biphasic conditions or spatial separation[31] of water. As expected, the addition of the hydrophobic co-solvents such as toluene and o-xylene yielded significantly enhanced enantioselectivity (up to 87% ee) (Fig. 1c, see also entries 9–11 and 18–20 in Supplementary Table 1). These hydrophobic solvents can sequester the transition states away from water, consequently suppressing the water-catalyzed non-enantioselective pathway[32]. To verify this, we performed the same reaction with less acidic D$_2$O instead of H$_2$O. No solvent D/H isotope effect on reaction rate and enantioselectivity was observed, strongly indicating that, under our on-water conditions, the interfacial hydrogen bonding or a proton transfer process between the aqueous phase and organic reactants is not involved in the transition state (Fig. 1c, see also entry 21 in Supplementary Table 1). If our on-water catalysis results from the general acid mechanism, then the reaction ought to be slower on less acidic D$_2$O[10,33]. Evidence for the hydrophobic hydration effects on increased enantioselectivity was also obtained using an anti-hydrophobic agent, lithium perchlorate (LiClO$_4$)[34,35]. In contrast to the results obtained from the addition of antichaotropic (increasing the hydrophobic effect) salts such as NaCl, the ee value decreased dramatically in aqueous LiClO$_4$ (15% ee for (R)-**3a** and 10% ee for (R)-**5a**) (Fig. 1c, see also entries 12 and 22 in Supplementary Table 1).

The strength of hydrophobic interactions can usually be controlled by introducing suitable hydrophobes in substrates and catalysts. Recently, we reported that catalyst hydrophobicity, which can affect hydrophobic interactions between catalysts and substrates, plays a critical role for the reaction rate in some on-water catalytic reactions[21,22]. Thus, we decided to examine the effect of the hydrophobicity of catalysts on enantioselectivity. Under the on-water reaction condition (entry 10 of Supplementary Table 1), the effect of the substituent at carbon C5 of several cinchonine-derived catalysts (**CN-1** to **CN-5**, Fig. 2a) on reaction

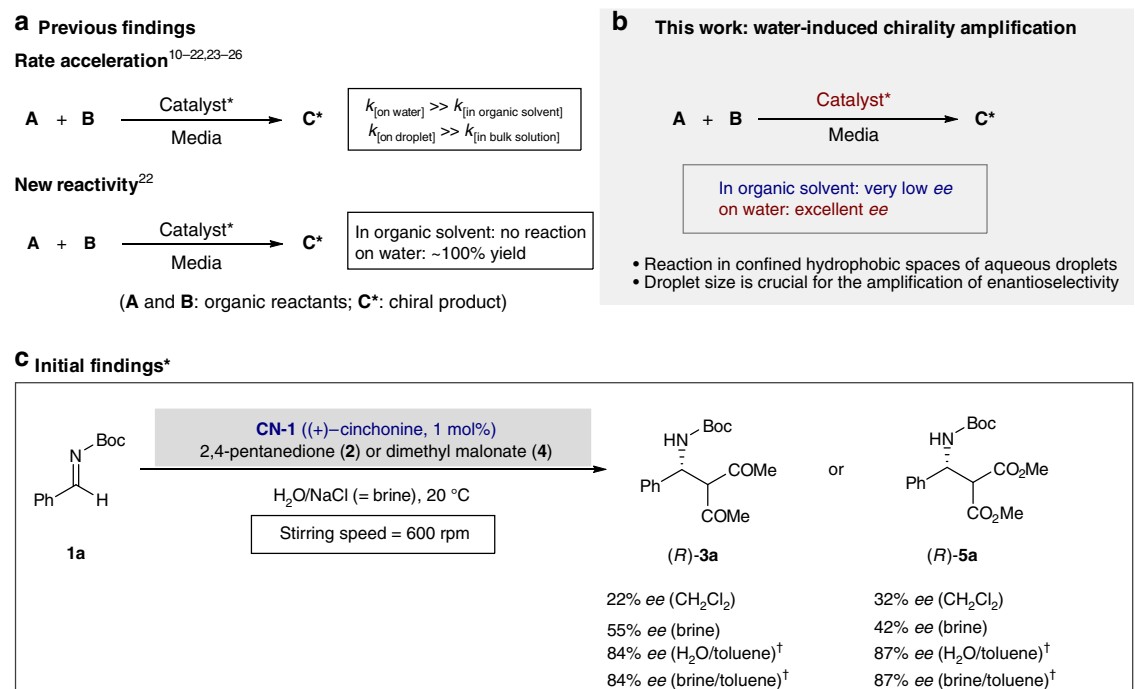

**Fig. 1** Concepts and initial findings on the water-induced hydrophobic amplification. **a** Previously observed rate acceleration under on-water and on-droplet conditions. **b** First observation of the chirality amplification under on-water conditions. **c** Mannich reaction with imine (**1a**) with 2,4-pentanedione (**2**) or dimethyl malonate (**4**) in various reaction media. *Conditions: Imine **1a** (0.3 mmol), 2,4-pentanedione **2** or dimethyl malonate **4** (0.6 mmol), and catalyst **CN-1** ((+)-cinchonine, 1 mol%) with 2.0 mL of reaction media at 20 °C for 1 h (for **2**) or 24 h (for **4**). †Toluene or *o*-xylene (15 equiv) was used as an additive. §See Supplementary Table 1 for more experimental results. *ee* = enantiomeric excess; Rpm = revolutions per minute; Boc = *tert*-butyloxycarbonyl

rate and enantioselectivity was investigated. Almost no effect of catalyst hydrophobicity on rate acceleration was observed. However, surprisingly, a significant hydrophobic chirality amplification (from 84% *ee* to 96% *ee*) derived from the catalyst substituent effect was observed in the catalytic Mannich reactions of **1a** with **2** (Fig. 2b). Relatively more hydrophobic catalysts exhibited higher catalytic enantioselectivities than the less hydrophobic catalysts. Thus, among the screened catalysts, **CN-5** (Log *P* = 7.34) gave the best result (96% *ee*). These trends were also observed using the pseudoenantiomeric cinchonidine-based catalysts (**CD-1** to **CD-5**: from 72% *ee* to 92% *ee*). In contrast, in CH₂Cl₂, regardless of the catalyst structure, much lower enantioselectivities (10–24% *ee*) were obtained. This result supports the assumption that the substituents at the C5 position of the catalysts only function as hydrophobic tags. Also in the reactions with dimethyl malonate (**4**), the hydrophobic group tagged **CN-2** and **CN-3** and their cinchonidine analogs showed slightly higher enantioselectivity than those obtained with other catalysts examined in this study. However, the effect of catalyst hydrophobicity on enantioselectivity was not significant like that observed in the reactions with acetylacetone (**2**) (see Supplementary Table 2).

**Stirring-induced chirality amplification**. It is well known that, under on-water conditions, the control of micelles by stirring rate can be critical for the outcome of chemical reactions[36,37]. Thus, the influence of the stirring rate on the catalytic results of the Mannich reaction of **1a**–**1c** with **2** or **4** was investigated by varying the stirring rate from 200 rpm to 1150 rpm (Fig. 2c). Surprisingly, the dramatic effect of the stirring rate on the enantioselectivity was observed under our on-water reaction

condition. The enantioselectivity of the Mannich reaction was increased by increasing the stirring speed. A maximum *ee* (97% *ee* for **3a** and 91% *ee* for **5a**) was then reached at a stirring speed of 1150 rpm while lower *ee*s were obtained at a lower stirring speed. These results indicate that the sizes of the droplets play a vital role in promoting stereoselectivity in this reaction. With increasing stirring rate, the size of the emulsion or suspension formed decreases; the hydrophobic organic solutes encapsulated inside transient microscopic hydration shell are consequently more strongly pressurized. Therefore, this might lead to an increased local concentration of the hydrophobic reactants, presumably forming a confined transition states. The catalytic-like effect of the stirring rate on enantioselectivity is expected to be of interest for both research and technological applications.

**Quantification of the on-water effect on enantioselectivity**. Although on-water conditions in this study enabled a dramatic increase in enantioselectivity, precise control of the droplet size was not amenable in a reaction flask due to many different factors[38]. Moreover, our attempts for in situ measurements of the droplet sizes in the emulsions during the reaction was not successful, preventing any quantitative analysis of the effect of droplet size on the enantioselectivity. Therefore, to quantify the on-water effect on enantioselectivity, and, thus, to better understand the mechanism underlying this water-induced chirality amplification, we employed a biphasic microfluidic system to generate precisely defined monodisperse water and organic phase plugs.

The Mannich reaction of **1a** with **4** using 1 mol% of catalyst **CN-2** or **CD-2** as a model reaction was performed in a biphasic microfluidic system (*o*-xylene/brine). A biphasic microfluidic system was set using a cross-junction meeting of controlled flows

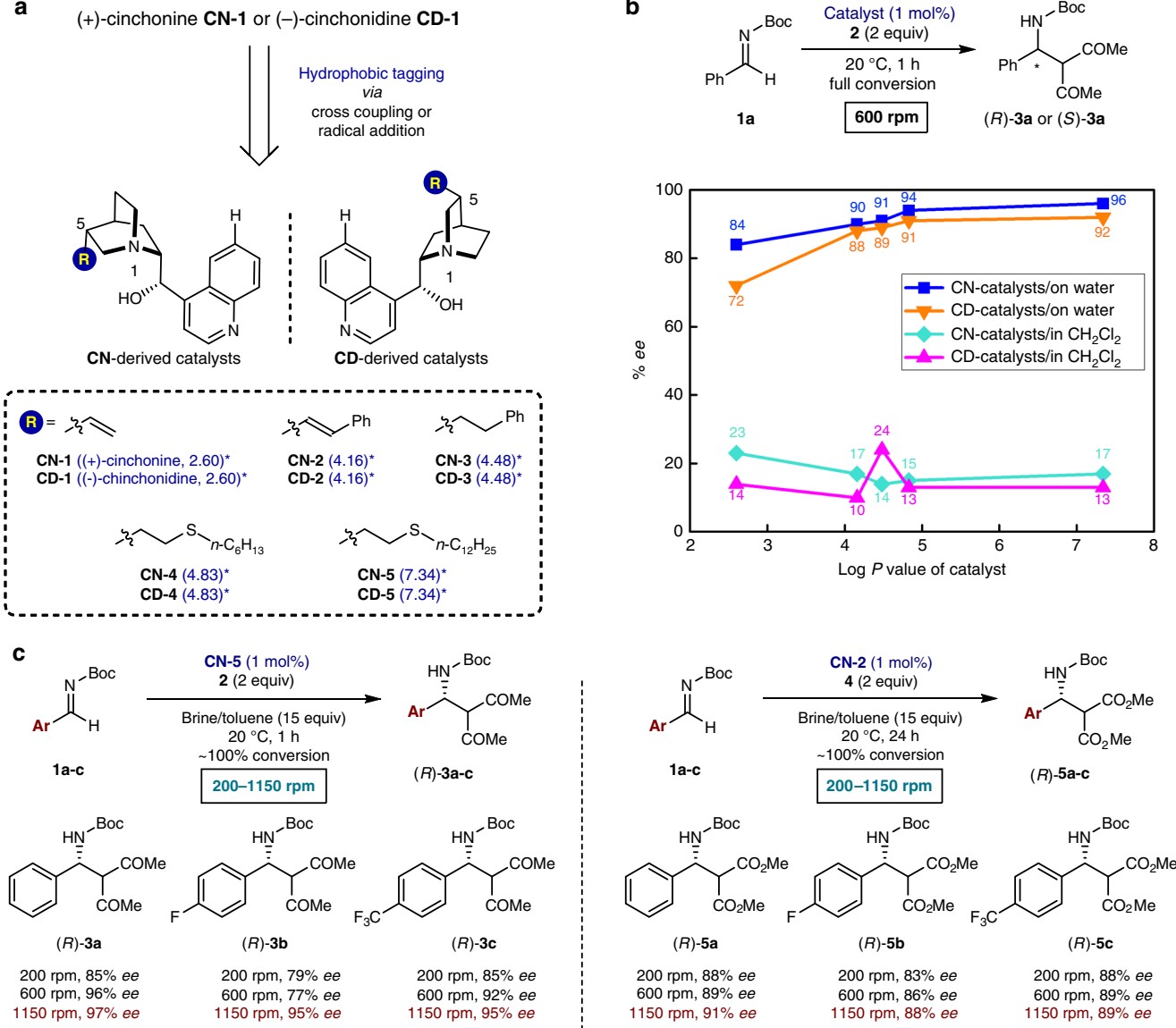

**Fig. 2** Enantioselectivity of the Mannich product depending on catalyst hydrophobicity and stirring rate. **a** Chiral catalysts used in this study. **b** Effect of catalyst hydrophobicity on enantioselectivity. Conditions: **1a** (0.3 mmol), **2** (0.6 mmol), catalyst (1 mol%), and brine (2.0 mL)/toluene (15 equiv) or CH$_2$Cl$_2$ (2.0 mL). **c** Effect of stirring speed on enantioselectivity. Conditions: **1a**–**1c** (0.3 mmol), **2** or **4** (0.6 mmol), catalyst (1 mol%), and brine (2.0 mL)/toluene (15 equiv). For experimental details regarding controlling the stirring speed, see the Supplementary Figure 1. Enantiomeric excess (% *ee*) was determined by high-performance liquid chromatography (HPLC). *The Log *P* values (the *P* value is a *n*-octanol/water partition coefficient) of catalysts were calculated using ChemBioDraw Ultra 12.0 software

of brine and organic solutions containing the organic reactants and catalyst **CN-2** or **CD-2**. The reaction mixture was injected into the system using a syringe pump, collected at the outlet of the fluorinated ethylene propylene (FEP) tubing (inner diameter (ø) = 250, 500, and 1000 μm, length = 15 m) (Fig. 3a). A series of droplets with different sizes was formed in the FEP tubing by adjusting the flow rate ratio (Q$_w$/Q$_o$) between the aqueous and organic phases (Fig. 3b). After the tube was filled with the reaction mixture, the flows of brine and organic solutions were stopped, and the outlet of the tube was then sealed with a plastic paraffin film. The other end of the tube was also sealed with a plastic paraffin film. The static biphasic plugs were then kept inside the FEP tubing for 48 h at 20 °C. Without any stirring, the reaction proceeded smoothly even in the static droplets. Furthermore, as expected, the enantioselectivity was significantly dependent on the droplet size (Fig. 3c and Supplementary

Table 3). As the size of the droplet decreased, the *ee* values increased. When using the same tube, the *ee* values of the product increased when the Q$_w$/Q$_o$ increased. Under the same brine–organic flow ratios (Q$_w$/Q$_o$), the smaller size of the inner diameter of the microtube gave the higher *ee* values. Furthermore, the employment of a biphasic microfluidic system afforded higher *ee* values, exceeding the maximum *ee*s obtained under on-water batch conditions (92.5% *ee* vs. 91% *ee* using **CN-2** catalyst; 92% *ee* vs. 85% *ee* using **CD-2** catalyst) (Fig. 3c and Fig. 4, see also Supplementary Table 3). However, no chirality amplification was observed when the organic plugs were generated by injecting argon gas instead of water into the chip reactor, regardless of sizes of organic droplets, further confirming the crucial role of water for the chirality amplification. However, interestingly, the droplet size showed almost no effect on the reaction kinetics (see Supplementary Table 4 for the detailed kinetic data).

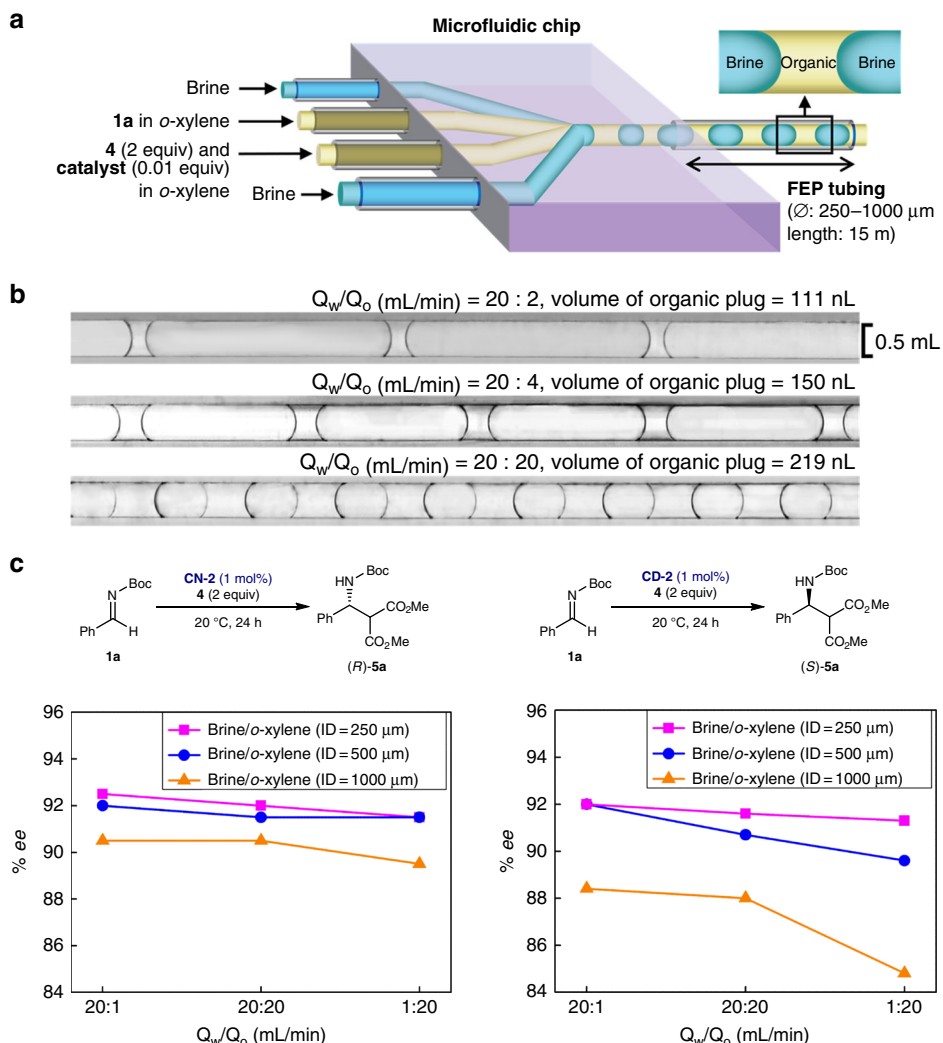

**Fig. 3** Microfluidic experiments. **a** Flow microreactor system for the Mannich reaction of **1a** with **4** using 1 mol% of **CN-2** or **CD-2**: plugs form at the junction between a o-xylene solution containing the reagents (organic phase) and an aqueous phase (brine), and travel down the FEP (fluorinated ethylene propylene) tubing where the reaction occurs. **b** Plug volume can be controlled by varying the relative flow rates of the two phases (Q$_w$: aqueous flow rate, Q$_o$: organic flow rate). *Plug volume was determined by dividing the organic flow rate (Q$_o$) by the droplet production frequency, measured using a high speed camera. **c** Effect of biphasic microfluidic conditions (i.e., droplet size) on enantioselectivity

It is important to note here that, to date, the beneficial confinement effect on enantioselectivity has only been observed when the system is confined at molecular scales (pore size < 2 nm)[2]. However, in this study, the enantioselectivity could be enhanced by confinement at a microscale, in droplets with diameters in the micrometer range. As mentioned previously, we presume that the water-induced chirality amplification can be ascribed to hydrophobic hydration effect. According to the recent spectroscopic and calculation studies, the origin of hydrophobic hydration is the strengthened water hydrogen bonding near purely hydrophobic solutes[4], i.e., in the presence of hydrophobic reactants, the water forms a transient microscopic hydration shell which shows extensive structural ordering resembling that in ice and clathrates. Thus, we presume that this strong H-bond network of water molecules can squeeze out the hydrophobic reactants[39]. This water-induced confinement effect can consequently enforce proximity of the catalyst and substrates within the confined hydrophobic cavities of micro-droplets and leads to a more compact transition state.

**Experiments for mechanism elucidation.** It is known that, under high pressure conditions, a more compact transition state (e.g., a negative volume of activation) is operative[1,16]. Thus, to determine whether water-induced enhancement of enantioselectivity can be ascribed to the hydrophobic hydration effect (i.e., whether water molecules involved in the strengthened hydrogen bonds pressurize the reactants), we first conducted the above-described biphasic microfluidic Mannich reaction of **1a** with **4** using 1 mol % of **CD-2** under high pressure. The pressure within microfluidic system (ø = 1000 μm, Q$_w$/Q$_o$ = 1/20) was controlled using a back pressure regulator. As anticipated, a noticeably higher enantioselectivity was observed even under ca. 5 bar (from 84% ee under 1 bar (Fig. 3c) to 90% ee under ca. 5 bar (Fig. 4a)). Next, we also performed the same reaction in an organic solvent (CH$_2$Cl$_2$) under 2 kbar (see Supplementary Method for the detailed experimental procedure)[40]. A remarkably higher enantioselectivity was observed under 2 kbar (89% ee for **1a** and 94% ee for **1d**). However, under atmospheric pressure and at the same reaction temperature, much lower enantioselectivities (55% ee for

**Fig. 4** High pressure experiments. **a** High pressure effect on enantioselectivity under biphasic microfluidic conditions using a BPR (back pressure regulator). †Reaction conditions: **1a** (1 equiv), **4** (2 equiv) and **CD-2** (0.01 equiv), ∅ = 1000 μm, $Q_w/Q_o$ = 1/20. **b** High pressure effect on catalytic Mannich reaction in an organic solvent. †Reaction conditions: **1a** (0.24 mmol), **4** (0.48 mmol) and catalyst (1 mol%) in $CH_2Cl_2$ (4.0 mL) at −20 °C. The melting point of water under 2 kbar is −20.8 °C[48]

**1a** and 48% *ee* for **1d**) were obtained (Fig. 4b). All the above-mentioned results (i.e., the inceased *ee* values under high pressure conditions) indicate that in the present reaction, the more enantioselective transition state might be more compact than a less enantioselective transition state. Furthermore, these results concur with those obtained from recent studies, which reported that phenomena occurring only at extremely high pressures in a bulk phase can be observed in a liquid confined within porous materials at significantly lower pressures. This implies that the pressure in such a confined phase is remarkably large[41,42].

**Generality of water-induced chirality amplification**. Flow and batch reactors were evaluated to confirm the generality of water-induced hydrophobic chirality amplification. The scope of the Mannich reaction of the diverse aromatic imines (**1a**–**1i**) with dimethyl malonate (**4**) was investigated using 1 mol% of catalyst **CN-2** or **CD-2**. Regardless of the steric and electronic nature of the aromatic ring, in all substrates tested in this study, a dramatic increase in enantioselectivity was observed under both on-water and biphasic microfluidic conditions (condition *b* and condition *c*-1, respectively, of Fig. 5) compared with those obtained in a homogeneous organic solvent (condition *a* of Fig. 5). For example, in the case of (*S*)-**5a**, 91% *ee* was obtained using a biphasic microfluidic system, while a homogeneous condition (in *o*-xylene) gave an almost racemic product. Notably, the biphasic microfluidic system always gave higher enantioselectivities than those obtained under on-water conditions, probably due to the smaller sizes of microfluidic droplets than those of on-water droplets. Furthermore, under the biphasic microfluidic condition, this high level of enantioselectivity could be maintained with much lower catalyst loading. Under the biphasic microfluidic condition, the reaction proceeded even with 0.2 mol% of catalyst

loading (condition *c*-2) without any erosion of enantioselectivity (Fig. 5), albeit longer reaction time was required. In contrast to these results, as described previously, 10 mol% of catalyst loading and very low reaction temperature (−35 °C) are usually required to achieve similar level of enantioselectivity in organic solvents.

## Discussion

In summary, we demonstrated that water can induce the chirality amplification in a catalytic asymmetric reaction. Under on-water conditions, the enantioselectivity of a catalytic reaction can be significantly enhanced in the confined hydrophobic spaces of organic droplets surrounded by water. More significantly, this chirality amplification can be further increased by decreasing the droplet size. The droplet size effect on the enantioselectivity was quantified by using the biphasic microfluidic technique. Although in-depth mechanistic studies are still needed in order to fully understand the role of water, from all the experimental results obtained in this study, we can conclude that this water-induced chirality amplification can be attributed to the hydrophobically induced confinement effect. This remarkable observation could provide some inspiration for developing new strategies to enhance enantioselectivity and thus has the potential to open a new chapter in the field of asymmetric catalysis. In addition, our discovery that the enantioselectivity can be greatly amplified in the confined cavities of water cages could help unlock secrets of homochirality which is a fundamental component of molecular recognition in biological systems[43–47]. Considering the aqueous environment of early Earth (mist, clouds, and spray, etc.), this scenario for chirality amplification which led to the present homochiral state, might be plausible. Furthermore, it is likely that the limited diffusion of enantio-enriched products in such

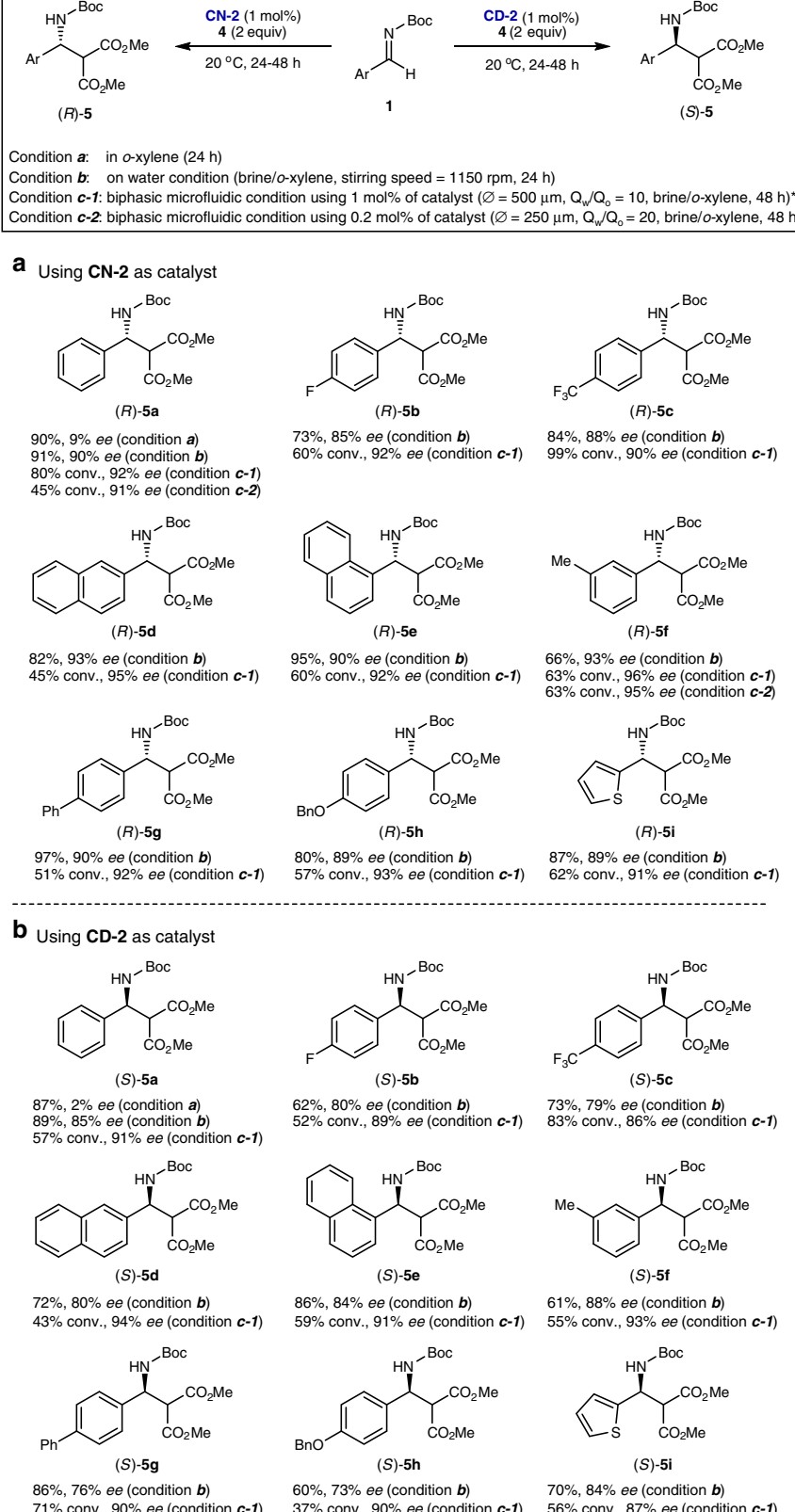

**Fig. 5** Substrate generality of water-induced hydrophobic chirality amplification in the Mannich reaction. **a** Using **CN-2** as catalyst. **b** Using **CD-2** as catalyst. *The isolated yields could not be determined since the total amount (~240 μL) of organic reaction mixture in the micro-tube was too small to determine the correct yield. We, thus, determined the conversion by ¹H NMR integration instead of the isolation yields. We assumed that the imine **1** can be converted to the corresponding product **5** since the nucleophile **4** was oversupplied (the molar ratio of **1:4** = 1:2), and the hydrolysis of **1** to aldehyde was negligible compared to the imine consumption by the reaction

droplets might provide a chance to participate in a self-replicating, evolvable system in the prebiotic era.

## Methods

**Asymmetric Mannich reactions under on-water conditions**. Acetylacetone **2** or dimethyl malonate **4** (0.6 mmol) was added to the mixture of *N*-Boc imine **1** (0.3 mmol), catalyst (0.003 mmol, 1 mol%), toluene or *o*-xylene (4.5 mmol), and water or brine (2.0 mL). The reaction mixture was stirred vigorously with a magnetic bar at 1150 rpm and the temperature was set at 20 °C. After completion of the reaction (0.5–1 h for **2** and 24–48 h for **4**, respectively), the reaction mixture was extracted with ethyl acetate (3 × 5 mL). The combined organic layers were washed with brine, dried over anhydrous sodium sulfate, filtered, and concentrated in vacuo. The residue was purified by column chromatography on silica gel eluting with a hexanes/ethyl acetate mixture, affording the corresponding Mannich product **3** or **5**.

**Asymmetric Mannich reactions under biphasic microfluidic conditions**. The organic solutions (solution 1: dimethyl malonate **4** (69 μL, 0.6 mmol) and catalyst **CN-2** or **CD-2** (1.1 mg, 0.003 mmol) in *o*-xylene (0.88 mL), the ultrasonification was used to assist the dissolution of catalyst in *o*-xylene; solution 2: phenyl *N*-Boc imine **1a** (61.6 mg, 0.3 mmol) in *o*-xylene (0.88 mL)) and the aqueous solution (brine, 5.13 M) were loaded into separate syringes, then injected onto the microfluidic chip at different flow rates. The brine and organic solutions were brought together on the chip, then coflowed and compartmentalized into droplets by flow focusing of the aqueous phase with an organic phase. After the tube was filled with the reaction mixture, the flows of brine and organic solutions were stopped, and the outlet of the tube was then sealed tightly with a plastic paraffin film. The other end of the tube was also sealed with a plastic paraffin film. The biphasic plugs were kept inside the FEP tubing at 20 °C without any shaking. After 24–48 h, the reaction mixture was collected from the outlet of the tubing by flushing with argon. The organic phase was then purified by chromatography to determine the *ee* values. The conversions of the reaction mixture were determined using ¹H NMR spectroscopy by comparing the product peak integration with the reactant peak integration.

**Characterization**. The % *ee* values of the Mannich products **3** and **5** were determined by chiral HPLC.

For NMR spectra of the synthesized compounds in this article, see Supplementary Figures 5–62. For HPLC spectra of the Mannich products **3** and **5**, see Supplementary Figures 72–248.

## Data availability

The authors declare that all relevant data supporting the findings of this study are available within the article and Supplementary Information files, and also are available from the corresponding author upon reasonable request.

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

## Acknowledgements

We are grateful for the financial support provided by the Ministry of Science, ICT and Future Planning (NRF-2016R1A4A1011451 and NRF-2017R1A2A1A05001214). The authors are also grateful to Dr. S. Paladhi (Sungkyunkwan University), Dr. Y.S. Lee (Sungkyunkwan University), Dr. J.H. Sim (Sungkyunkwan University), and Prof. J.W. Lee (University of Kopenhagen) for their assistance in the experiments and for their helpful discussions.

## Author contributions

C.E.S. designed the research. I.-S.H., S.J.P., M.J.J., S.Y.S., H.Y.B. and J.Y.J. performed experiments and carried out the analysis. H.Y.B. and J.Y.J. contributed to the initial studies. S.J.P., I.-S.H., M.J.J. and S.Y.S. performed the optimization studies on reaction conditions. S.J.P. and I.-S.H. contributed to microfluidic studies. S.J.P. and I.-S.H. contributed equally to this work. C.E.S. directed and supervised the project. C.E.S. wrote the paper.

## Additional information

**Competing interests:** The authors declare no competing interests.

