## [Peer Review File · Nature Communications]

Reviewers' comments:

Reviewer #1 (Remarks to the Author):

I believe that this manuscript does present new, interesting experimental results but the interpretation of the findings makes me uncomfortable about recommending publication in its present form. I suggest a major revision is needed and then the manuscript will be need to be considered again.

This paper examines the effect of adding brine on the enantioselectivity of a Mannich reaction between tert-butyl (E)-benzylidenecarbamate and dimethyl malonate or 2,4-pentanedione catalyzed by cinchonine or cinchonidine derivatives. This reaction was previously reported in dichloromethane and very low temperatures and high catalyst loadings (10%) were necessary to achieve enantioselectivity. The authors find that with the addition of brine, the reaction proceeds at room temperature with good enantioselectivities at lower catalyst loadings (1%). This is news and worthy of publication.

The authors attribute this to a "water induced hydrophobic chirality amplification", which is essentially a hydrophobic effect; when hydrophobic reagents are forced closer together, the authors postulate that the transition state becomes "more compact" allowing for an increase in enantioselectivity. They examine the effects of various parameters of this reaction, including stir rate, hydrophobicity of the organocatalyst, pressure, droplet size (using a microfluidic system), and explore the scope of this reaction. In all cases, they find that enantioselectivities increase with decreasing droplet sizes. To me, however, this discussion is the weakest aspect of this manuscript.

Points to consider:

I disagree with the authors' characterization that this is an "on water" reaction. Both dimethylmalonate and 2,4-pentanedione are highly soluble in water, suggesting that this is actually an "in water" process. For a true on-water reaction, reagents should be insoluble. I do not know the reason for the increase in enantioselectivity, but it is likely that there is an interfacial effect or a complex hydrogen bonding interaction between substrates and cinchona catalysts that occurs once water is added that allows for enantioselective induction. The transition state of a Mannich reaction is quite polar; I doubt the reagents are confined surrounded by a shell of water. If this claim is made,

then I feel that some experimental evidence needs to be offered in support of this contention. Rather, I believe that water might be an active participant in the process. For a distinction between "in water" and "on water" chemistry, please see R. N. Butler, A. G. Coyne, *Org. Biomol. Chem.* 2016, 14, 9945 – 9960.

2. I find the statement on pg. 6, lines 175-177 surprising: 'Although the droplet size has almost no effect on the reaction kinetics, the enantioselectivity was significantly dependent on the droplet size.' People disagree about why reactions are accelerated in confined environments and about the magnitude of acceleration; however, it has been demonstrated using a variety of methods that droplet size generally has an inverse relationship to reaction rate.

Reviewer #2 (Remarks to the Author):

CE Song et al. *Nature* Chirality Amplification in confined water cages

This high quality paper describes extensive experiments and is well written. The subject matter will be of great interest to almost all chemists. The major effect reported, the amplification of enantiomeric selectivity, is indeed remarkable.

That said, there are also some questions which the authors should address.

- i) The evidence that the effect results from 'hydrophobic hydration' is not completely compelling. Contributions to selective stabilization of one over the other form of the transition state through effects like cation stabilization of one configuration over the other at the interface, e.g. Na⁺ stabilization of one configuration through crown-like binding, might also be considered. The assumption is made that the catalyst and substrates are brought together in the transition state but there is no evidence regarding transition state structure.
- ii) Do the authors have information on what would happen if the on-water conditions were to be attempted using water rather than brine solution with organic solvents?
- iii) Derived from question 2, ions of alkali metals easily form adducts with coordination of oxygen in ketone and ester because they are hard Lewis acids and bases, could this be a reason for

the chiral selectivity? Perhaps the authors should also try some soft Lewis acid like Ag(I) to form the brine solution to see if there is still chiral selectivity.

iv) The abstract mentions 'chirality amplification' which is clearly a feature of the experiments as well as 'chiral transmission' the meaning of which is less clear in this context. If this refers to the fact that a chiral catalyst is used in the experiments then the term is correctly used although it is not necessary and might even be confusing.

v) The observed effect on enantioselectivity is described in several places as 'water induced' this implies a water cage which seems reasonable but again there is little direct evidence for this, e.g. from computation. The fact that 'hydrophobic hydration' can accelerate reactions has previously been reported [15 – 22] so there is certainly precedent for effects on the transition state.

vi) It is noteworthy that in the present study, rate acceleration is not reported. The authors should offer an explanation as to how changes in the transition state can cause chiral amplification without any changes in reaction rate. (e.g. page 6 second paragraph "although the droplet size has almost no effect on the reaction kinetics, the enantioselectivity was significantly dependent on the droplet size")

vii) Enantiomers can have different densities so the high-pressure experiment is not sufficient to prove the compact transition state structure.

The suggestion that the data might have implications for the chemical evolution of homochirality is interesting but the use of a chiral organic catalyst in this study simply pushes the question back a step: how did the chiral catalyst evolve its homochirality? The references cited in the discussion of this step [41- 43] do not address this question. More pertinent perhaps are studies in which an increase in ee is observed without a catalyst, starting from the slight natural (avoidance of weak nuclear force parity violation) excess of one enantiomer over another, see *Angew. Chem. Int. Ed.*, 45 (2006) 554-569 and *J. Phys. Chem. A* (2015), 119, 12805–12822.

The droplet size effect on chiral amplification is particularly interesting because there is a strong size effect on reaction acceleration in droplets, cited in the review [23] but more thoroughly demonstrated in subsequent work from the same groups of authors. This extends the parallel between the droplet confined volume and the on-water confined conditions of the present study but it also underlines the question of how chiral amplification can be observed without apparent rate effects in the present study.

Reviewer #1 (Remarks to the Author):

I believe that this manuscript does present new, interesting experimental results but the interpretation of the findings makes me uncomfortable about recommending publication in its present form. I suggest a major revision is needed and then the manuscript will be need to be considered again.

This paper examines the effect of adding brine on the enantioselectivity of a Mannich reaction between tert-butyl (E)-benzylidenecarbamate and dimethyl malonate or 2,4-pentanedione catalyzed by cinchonine or cinchonidine derivatives. This reaction was previously reported in dichloromethane and very low temperatures and high catalyst loadings (10%) were necessary to achieve enantioselectivity. The authors find that with the addition of brine, the reaction proceeds at room temperature with good enantioselectivities at lower catalyst loadings (1%). This is news and worthy of publication.

Response:

To our delight, we further observed that the high level of enantioselectivity could be maintained with much lower catalyst loading. Under the biphasic microfluidic condition, the reaction proceeded smoothly even with 0.2 mol% of catalyst loading (condition c-2 of Fig. 5) without any significant loss of enantioselectivity (91% ee for **5a** and 95% ee for **5f**), albeit longer reaction time was required. In contrast to these results, 10 mol% of catalyst loading and very low reaction temperature (-35 °C) are usually required to achieve similar level of enantioselectivity in organic solvents.

To the best of our knowledge, this is the lowest catalyst loading for organocatalytic Mannich reactions reported to date, while achieving excellent enantioselectivities. These results were included in the revised manuscript (condition c-2 of Fig. 5) and Supplementary Information (S117 and S123).

The authors attribute this to a "water induced hydrophobic chirality amplification", which is essentially a hydrophobic effect; when hydrophobic reagents are forced closer together, the authors postulate that the transition state becomes "more compact" allowing for an increase in enantioselectivity. They examine the effects of various parameters of this reaction, including stir rate, hydrophobicity of the

organocatalyst, pressure, droplet size (using a microfluidic system), and explore the scope of this reaction. In all cases, they find that enantioselectivities increase with decreasing droplet sizes. To me, however, this discussion is the weakest aspect of this manuscript.

Response:

Thank you for your comments. As you pointed out, we made a very important discovery that water can act as a kind of chirality amplifier, i.e., the enantioselectivity can be greatly amplified in the confined cavities of water cages. We carefully revised to clarify our perspective on the experimental results.

Points to consider:

I disagree with the authors' characterization that this is an "on water" reaction. Both dimethylmalonate and 2,4-pentanedione are highly soluble in water, suggesting that this is actually an "in water" process.

For a true on-water reaction, reagents should be insoluble. I do not know the reason for the increase in enantioselectivity, but it is likely that there is an interfacial effect or a complex hydrogen bonding interaction between substrates and cinchona catalysts that occurs once water is added that allows for enantioselective induction. The transition state of a Mannich reaction is quite polar; I doubt the reagents are confined surrounded by a shell of water. If this claim is made, then I feel that some experimental evidence needs to be offered in support of this contention. Rather, I believe that water might be an active participant in the process. For a distinction between "in water" and "on water" chemistry, please see R. N. Butler, A. G. Coyne, *Org. Biomol. Chem.* 2016,14, 9945 – 9960.

Response:

We believe that our system is typical "on water" rather than "in water" due to the following reasons.

1. The substrates, acetyl acetone and dimethyl malonate, are slightly soluble in water. However, under our condition (e.g., using a hydrophobic co-solvent such as toluene or *o*-xylene), acetyl acetone and dimethyl malonate are insoluble in water or in brine, as can be seen in Supplementary Figure 2. Even without a hydrophobic co-solvent, these compounds are insoluble in brine (see the Figure below). It is understood that the organic phase is dispersed, not solubilized, therefore demonstrating an on-water system.

Supplementary Figure 2. Pictures of asymmetric Mannich reaction under on-water conditions. a. before reaction; b. during reaction; c. after reaction.

a. acetyl acetone dispersed on brine; b. dimethyl malonate dispersed on brine

2. To provide further evidence, we also examined the Mannich reaction with more hydrophobic diethyl malonate of which the solubility in brine is negligible (i.e., on-water conditions). Also in this case, a strong water-induced chirality amplification was observed (see the Scheme below).

3. We used brine where NaCl decreases the solubility of organic species in water by salting out, consequently resulting in increased hydrophobic effect. In contrast to the results obtained from the addition of NaCl, the ee value decreased dramatically using an anti-hydrophobic (or salting-in) agent such as LiClO₄. Please see Fig. 1 of the revised manuscript and Supplementary Table 1 (15% ee for **3a**; 10% ee for **5a**).

I find the statement on pg. 6, lines 175-177 surprising: 'Although the droplet size has almost no effect on the reaction kinetics, the enantioselectivity was significantly dependent on the droplet size.' People disagree about why reactions are accelerated in confined environments and about the magnitude of acceleration; however, it has been demonstrated using a variety of methods that droplet size generally has an inverse relationship to reaction rate.

Response:

In our study, we discovered that the enantioselectivity can be greatly amplified by the on-water system. Furthermore, this amplification is significantly dependent on droplet size in the continuous flow demonstration. It is understood that, as you commented, a size of droplets alters reaction kinetics as cited in a review [23]. However, as you can see from our kinetic results (Supplementary Table 4), the droplet size has no noticeable effects on the reaction rate of the current enantioselective catalysis.

Supplementary Table 4. The effect of droplet sizes on conversion in the biphasic microfluidic reaction conditions. Note that almost no change was observed by varying the size of droplets.

This is also puzzling for us, however, we might explain with a possible hypothesis: the rate determining step is dissociated from the enantio-determining step.

There are precedents where the rate-determining step (RDS) of the reaction is dissociated from the enantio-determine step (EDS). In this case, in order to explain the increased enantioselectivity, the rate determining step does not need to be considered. Further in-depth kinetic analysis would be required to determine the RDS and EDS which will be investigated in due course.

We would like to mention one important example here. As described in our manuscript, a significant improvement in the enantioselectivity of a chiral catalyst could be achieved by taking advantage of confinement effects of nanopspaces of porous materials [ref. 2] The increase of the ee values is usually observed inversely proportional to the size of nano cages. However, also in these cases, several examples have been reported **where the size of the nanocages has almost no effect on the reaction rate**, albeit the apparent differences of diffusion rates of molecules to the catalytically active center. One example is that the Rh(COD)- and Pd(allyl)-complex of AEP anchored on MCM-41, **A** and **B**, gave 80% ee and 87% ee, respectively, in the hydrogenation of methyl benzoyl formate, while the corresponding homogeneous catalysts gave a racemic product. However, as you can see from the scheme below, the nanocage has almost no effects on the reaction rate of this enantioselective reaction (Ref: *Angew. Chem. Int. Ed.* **2003**, *42*, 4326-4331).

Using Rh(COD)AEP; 95% conversion after 24 h, 0% ee
Using heterogeneous catalyst **A**; 99% conversion after 24 h, 80% ee

Using Pd(allyl)AEP; 100% conversion after 2 h, 0% ee
Using heterogeneous catalyst **B**; 97% conversion after 2 h, 87% ee

Reviewer #2 (Remarks to the Author):

CE Song et al. Nature Chirality Amplification in confined water cages

This high quality paper describes extensive experiments and is well written. The subject matter will be of great interest to almost all chemists. The major effect reported, the amplification of enantiomeric selectivity, is indeed remarkable.

Response:

We appreciate your highly positive comments.

That said, there are also some questions which the authors should address.

i) The evidence that the effect results from 'hydrophobic hydration' is not completely compelling. Contributions to selective stabilization of one over the other form of the transition state through effects like cation stabilization of one configuration over the other at the interface, e.g. Na⁺ stabilization of one configuration through crown-like binding, might also be considered. The assumption is made that the catalyst and substrates are brought together in the transition state but there is no evidence regarding transition state structure.

Do the authors have information on what would happen if the on-water conditions were to be attempted using water rather than brine solution with organic solvents?

Response:

It is well known that NaCl decreases the solubility of organic species in water by salting out, i.e. NaCl is a typical "antichaotropic" salt. Thus, we just used brine to maximize the hydrophobic effect.

According to your suggestions, we carried out the Mannich reaction using pure water instead of brine. As you can see from the following schemes, identical enantioselectivities were obtained with those obtained with brine. Only a slight difference in rates was observed. The reaction using brine proceeded slightly faster than using pure water (Please see the Figure below). These results were included in the revised manuscript. The full data were also added in Supplementary Table 1 and Supplementary Table 3.

The effect of reaction conditions (H₂O vs brine) on enantioselectivity

a. Under on-water condition

b. Under microfluidic condition

The effect of reaction conditions (H₂O vs brine) on conversion

	conversion at 6 h [%]	conversion at 12 h [%]	conversion at 24 h [%]
on brine	54	82	>99
on pure water	49	66	90

iii) Derived from question 2, ions of alkali metals easily form adducts with coordination of oxygen in ketone and ester because they are hard Lewis acids and bases, could this be a reason for the chiral selectivity? Perhaps the authors should also try some soft Lewis acid like Ag(I) to form the brine solution to see if there is still chiral selectivity.

Response:

According to your suggestion, we examined the following Mannich reaction in saturated aqueous solution of AgNO₃. However, silver cation acted as a Lewis acid and, thus, promoted the rapid hydrolysis of the imine substrate **1a**. Thus, no Mannich product could be obtained.

iv) The abstract mentions 'chirality amplification' which is clearly a feature of the experiments as well as 'chiral transmission' the meaning of which is less clear in this context. If this refers to the fact that a chiral catalyst is used in the experiments then

the term is correctly used although it is not necessary and might even be confusing.

Response:

The abstract was modified accordingly.

v) The observed effect on enantioselectivity is described in several places as ‘water induced’ this implies a water cage which seems reasonable but again there is little direct evidence for this, e.g. from computation. The fact that ‘hydrophobic hydration’ can accelerate reactions has previously been reported [15 – 22] so there is certainly precedent for effects on the transition state.

Response:

As described previously, we discovered for the first time that water can be used as chirality amplifier. To explain the mechanism underlying this interesting discovery, we discussed with some of the leading experts for computational chemistry. Unfortunately, however, it was concluded that there is no calculation tool for this new observation to date.

vi) It is noteworthy that in the present study, rate acceleration is not reported. The authors should offer an explanation as to how changes in the transition state can cause chiral amplification without any changes in reaction rate. (e.g. page 6 second paragraph “although the droplet size has almost no effect on the reaction kinetics, the enantioselectivity was significantly dependent on the droplet size”)

Response:

In our study, we discovered that the enantioselectivity can be greatly amplified by the on-water system. Furthermore, this amplification is significantly dependent on droplet size in the continuous flow demonstration. It is understood that, as you commented, a size of droplets alters reaction kinetics as cited in a review [23]. However, as you can see from our kinetic results (Supplementary Table 4), the droplet size has no noticeable effects on the reaction rate of the current enantioselective catalysis.

Supplementary Table 4. The effect of droplet sizes on conversion in the biphasic microfluidic reaction conditions. Note that almost no change was observed by varying the size of droplets.

This is also puzzling for us, however, we might explain with one possible hypothesis: the rate determining step is dissociated from the enantio-determining step.

There are precedents where the rate-determining step (RDS) of the reaction is dissociated from the enantio-determine step (EDS). In this case, in order to explain the increased enantioselectivity, the rate determining step does not need to be considered. Further in-depth kinetic analysis would be required to determine the RDS and EDS which will be investigated in due course.

We would like to mention one important example here. As described in our manuscript, a significant improvement in the enantioselectivity of a chiral catalyst could be achieved by taking advantage of confinement effects of nanospaces of porous materials [ref. 2] The increase of the ee values is usually observed inversely proportional to the size of nano cages. However, also in these cases, several examples have been reported **where the size of the nanocages has almost no effect on the reaction rate**, albeit the apparent differences of diffusion rates of molecules to the catalytically active center. One example is that the Rh(COD)- and Pd(allyl)-complex of AEP anchored on MCM-41, **A** and **B**, gave 80% ee and 87% ee,

respectively, in the hydrogenation of methyl benzoyl formate, while the corresponding homogeneous catalysts gave a racemic product. However, as you can see from the scheme below, the nanocage has almost no effects on the reaction rate of this enantioselective reaction (Ref: *Angew. Chem. Int. Ed.* **2003**, *42*, 4326-4331).

Using Rh(COD)AEP; 95% conversion after 24 h, 0% ee
Using heterogeneous catalyst **A**; 99% conversion after 24 h, 80% ee

Using Pd(allyl)AEP; 100% conversion after 2 h, 0% ee
Using heterogeneous catalyst **B**; 97% conversion after 2 h, 87% ee

vii) Enantiomers can have different densities so the high-pressure experiment is not sufficient to prove the compact transition state structure.

Response:

Thank you for your comments. Although a chiral molecule and its enantiomer have the same density, it is potentially possible to have a different density in the presence of another chiral components (a chiral catalyst or homo/heterodimers of enantiomers). In our experiments, relatively very low catalyst loadings (up to 0.2 mol%) were used, therefore, we can exclude the effect of the chiral catalyst on the density. Also, it is known that, under high pressure conditions, a more compact transition state (e.g., a negative volume of activation) is operative. Thus, the high-pressure experiments can be usually employed to prove the compact transition state structure (Ref. [1,16]).

The suggestion that the data might have implications for the chemical evolution of homochirality is interesting but the use of a chiral organic catalyst in this study simply pushes the question back a step: how did the chiral catalyst evolve its homochirality? The references cited in the discussion of this step [41- 43] do not address this

question. More pertinent perhaps are studies in which an increase in ee is observed without a catalyst, starting from the slight natural (avoidance of weak nuclear force parity violation) excess of one enantiomer over another, see *Angew. Chem. Int. Ed.*, 45 (2006) 554-569 and *J. Phys. Chem. A* (2015), 119, 12805–12822.

Response:

Thank you for your kind comments. According to your suggestion, we cited one of the above-mentioned literatures in our revised manuscript (Ref. 43).

New reference for [43]: *Angew. Chem. Int. Ed.*, 45 (2006) 554-569.

Our finding is that the enantioselectivity can be greatly amplified in the confined cavities of water cages. As described in the manuscript, considering that the aqueous environment of early Earth resembled aerosol droplets (mist, clouds, and spray, etc.) at the surface of oceans, we believe that our results would offer one of the reasonable scenarios for the chirality amplification event in the prebiotic era.

The droplet size effect on chiral amplification is particularly interesting because there is a strong size effect on reaction acceleration in droplets, cited in the review [23] but more thoroughly demonstrated in subsequent work from the same groups of authors. This extends the parallel between the droplet confined volume and the on-water confined conditions of the present study but it also underlines the question of how chiral amplification can be observed without apparent rate effects in the present study.

Response:

In our study, we discovered that the enantioselectivity can be greatly amplified by the on-water system. Furthermore, this amplification is significantly dependent on droplet size in the continuous flow demonstration. It is understood that, as you commented, a size of droplets alters reaction kinetics as cited in a review [23]. However, as you can see from our kinetic results (Supplementary Table 4), the droplet size has no noticeable effects on the reaction rate of the current enantioselective catalysis.

Supplementary Table 4. The effect of droplet sizes on conversion in the biphasic microfluidic reaction conditions. Note that almost no change was observed by varying the size of droplets.

This is also puzzling for us, however, we might explain with a possible hypothesis: the rate determining step is dissociated from the enantio-determining step.

There are precedents where the rate-determining step (RDS) of the reaction is dissociated from the enantio-determine step (EDS). In this case, in order to explain the increased enantioselectivity, the rate determining step does not need to be considered. Further in-depth kinetic analysis would be required to determine the RDS and EDS which will be investigated in due course.

We would like to mention one important example here. As described in our manuscript, a significant improvement in the enantioselectivity of a chiral catalyst could be achieved by taking advantage of confinement effects of nanospaces of porous materials [ref. 2] The increase of the ee values is usually observed inversely proportional to the size of nano cages. However, also in these cases, several examples have been reported **where the size of the nanocages has almost no effect on the reaction rate**, albeit the apparent differences of diffusion rates of molecules to the catalytically active center. One example is that the Rh(COD)- and Pd(allyl)-complex of AEP anchored on MCM-41, **A** and **B**, gave 80% ee and 87% ee, respectively, in the hydrogenation of methyl benzoyl formate, while the

corresponding homogeneous catalysts gave a racemic product. However, as you can see from the scheme below, the nanocage has almost no effects on the reaction rate of this enantioselective reaction (Ref: *Angew. Chem. Int. Ed.* **2003**, *42*, 4326-4331).

Using Rh(COD)AEP; 95% conversion after 24 h, 0% ee
 Using heterogeneous catalyst **A**; 99% conversion after 24 h, 80% ee

Using Pd(allyl)AEP; 100% conversion after 2 h, 0% ee
 Using heterogeneous catalyst **B**; 97% conversion after 2 h, 87% ee

REVIEWERS' COMMENTS:

Reviewer #1 (Remarks to the Author):

Thank you for this additional information, which has helped me understand better this highly interesting reaction system of yours. I enthusiastically recommend publication.

Sincerely yours,

Richard N. Zare

Reviewer #2 (Remarks to the Author):

As originally submitted this was a remarkable study. The authors have addressed the reviewers' remarks and even though I still find aspects of their mechanistic suggestions difficult to accept, the quality of the original data plus the new experiments mean that the work should be published and authors should be permitted the authorial privilege of postulating mechanisms that they think best fit their data. My suggestions below are for the authors to consider and then to decide whether to make changes. My formal recommendation as a reviewer is that the paper be accepted as is; with whatever changes the authors may/or may not wish to make.

To the Authors:

1. Please consider again the title and especially the words "in Confined Water Cages". The on-water vs. in-water arguments are satisfactory but there is not real evidence for Confined Water Cages. The three word title "Hydrophobic Chirality Amplification" would be more compelling.
2. P.3 Improve 'Any solvent D/H isotope effect on reaction rate was not observed,' by saying 'No solvent D/H isotope effect on reaction rate was observed'
3. P.3. Improve "the interfacial hydrogen bonding or a proton transfer process between the aqueous phase and organic reactants does not involved in the transition state" by saying "the interfacial hydrogen bonding or a proton transfer process between the aqueous phase and organic reactants does not involve the transition state"
4. The pressure effects to me are not 'remarkable'. The authors state (p.8) that "remarkably higher enantioselectivity was observed under high pressure conditions (from 84% ee under 1 bar (Fig. 3c) to 90% ee" Elsewhere (1st response in author rebuttal) they argue that a rather similar change is insignificant "without any significant loss of enantioselectivity (91% ee for 5a and 95% ee for 5f)"

5. The most important change the authors have made is the additional sentence in the Conclusion. . Although in-depth mechanistic studies are still needed in order to fully understand the role of water, from all the experimental results obtained in this study, we can conclude that this water-induced chirality amplification can be attributed to the hydrophobically induced confinement effect.

6. I suppose the authors will not do this but the greatest weakness of this study is that it is described, from the title through the abstract and onwards, in terms of a hypothesis (hydrophobic confinement) for which the evidence is weak. By contrast, the fact of chiral induction speaks for itself. Rewriting and keeping the hypothesis late in the manuscript would make it even better.

Point-by-point responses to referees

REVIEWERS' COMMENTS:

Reviewer #1 (Remarks to the Author):

Thank you for this additional information, which has helped me understand better this highly interesting reaction system of yours. I enthusiastically recommend publication.

Sincerely yours,
Richard N. Zare

Response: We appreciate this assessment.

Reviewer #2 (Remarks to the Author):

As originally submitted this was a remarkable study. The authors have addressed the reviewers' remarks and even though I still find aspects of their mechanistic suggestions difficult to accept, the quality of the original data plus the new experiments mean that the work should be published and authors should be permitted the authorial privilege of postulating mechanisms that they think best fit their data. My suggestions below are for the authors to consider and then to decide whether to make changes. My formal recommendation as a reviewer is that the paper be accepted as is; with whatever changes the authors may/or may not wish to make.

Response: We appreciate this assessment.

To the Authors:

1. Please consider again the title and especially the words "in Confined Water Cages". The on-water vs. in-water arguments are satisfactory but there is not real evidence for Confined Water Cages. The three word title "Hydrophobic Chirality Amplification" would be more compelling.

Response: Thank you for your suggestion. However, it is well known that in "oil in water emulsion system (i.e., our system)", the water molecules form a network of hydrogen bonds around the hydrophobic molecules thereby forming water cages (i.e., hydration shell).

2. P.3 Improve 'Any solvent D/H isotope effect on reaction rate was not observed,' by saying 'No solvent D/H isotope effect on reaction rate was observed'

Response: According to the suggestion provided by Reviewer #2, this sentence was changed as follows.

"No solvent D/H isotope effect on the reaction rate was observed"

3. P.3. Improve “the interfacial hydrogen bonding or a proton transfer process between the aqueous phase and organic reactants does not involved in the transition state” by saying “the interfacial hydrogen bonding or a proton transfer process between the aqueous phase and organic reactants does not involve the transition state”

Response: According to the suggestion provided by Reviewer #2, we have changed the above sentence as follows.

“the interfacial hydrogen bonding or a proton transfer process between the aqueous phase and organic reactants is not involved in the transition state”

4. The pressure effects to me are not ‘remarkable’. The authors state (p.8) that “remarkably higher enantioselectivity was observed under high pressure conditions (from 84% ee under 1 bar (Fig. 3c) to 90% ee” Elsewhere (1st response in author rebuttal) they argue that a rather similar change is insignificant “without any significant loss of enantioselectivity (91% ee for 5a and 95% ee for 5f)”

Response: Thank you for your comments. Under high pressure conditions (2 kbar), the enantioselectivity was dramatically jumped from 55% ee to 89% ee for **5a** and from 48% ee to 94% ee for **5d** (Please see Figure 4b). Even under 5 bar (microfluidic condition), a noticeably higher enantioselectivity was observed under high pressure conditions (from 84% ee under 1 bar (Fig. 3c) to 90% ee (Fig. 4a). Thus, we have changed the above sentence as follows.

As anticipated, a noticeably higher enantioselectivity was observed even under 5 bar (from 84% ee under 1 bar (Fig. 3c) to 90% ee under ca. 5 bar (Fig. 4a)).

5. The most important change the authors have made is the additional sentence in the Conclusion. Although in-depth mechanistic studies are still needed in order to fully understand the role of water, from all the experimental results obtained in this study, we can conclude that this water-induced chirality amplification can be attributed to the hydrophobically induced confinement effect.

I suppose the authors will not do this but the greatest weakness of this study is that it is described, from the title through the abstract and onwards, in terms of a hypothesis (hydrophobic confinement) for which the evidence is weak. By contrast, the fact of chiral induction speaks for itself. Rewriting and keeping the hypothesis late in the manuscript would make it even better.

Response: Thank you for your very helpful comments. However, as we described in the conclusion part, we believe that, considering our experimental results and the known scientific discoveries (references 4, 39, 41 and 42), our hypothesis is reasonable. Nevertheless, we are trying to understand and explain the exact mechanism underlying this interesting finding. We hope that these results will be reported elsewhere in near future.